# Improved Sum of Residues Modular Multiplication Algorithm

Mohamad Ali Mehrabi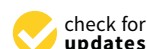

Department of Computing, Macquarie University, Sydney 2109, Australia; mohamadali.mehrabi@mq.edu.au

**Abstract:** Modular reduction of large values is a core operation in most common public-key cryptosystems that involves intensive computations in finite fields. Within such schemes, efficiency is a critical issue for the effectiveness of practical implementation of modular reduction. Recently, Residue Number Systems have drawn attention in cryptography application as they provide a good means for extreme long integer arithmetic and their carry-free operations make parallel implementation feasible. In this paper, we present an algorithm to calculate the precise value of "$X \mod p$" directly in the RNS representation of an integer. The pipe-lined, non-pipe-lined, and parallel hardware architectures are proposed and implemented on XILINX FPGAs.

**Keywords:** modular reduction; modular multiplication; residue number systems (RNS); Elliptic Curve Cryptography (ECC); sum of residues (SOR) reduction; montgomery modular reduction (MMR)

## 1. Introduction

The residue number system (RNS) has been proposed by Svoboda and Valach in 1955 [1] and independently by Garner in 1959 [2]. It uses a base of co-prime moduli $\{m_1, m_2, \cdots, m_N\}$ to split an integer $X$ into small integers $\{x_1, x_2, \cdots, x_N\}$ where $x_i$ is the residue of $X$ divided by $m_i$ denoted as $x_i = X \mod m_i$ or simply $x_i = \langle X \rangle_{m_i}$.

Conversion to RNS is straightforward. Reverse conversion is complex and uses the Chinese Remainder Theorem (CRT) [3]. Addition, subtraction, and multiplication in RNS are very efficient. These operations are performed on residues in parallel and independently, without carry propagation between them. The natural parallelism and carry-free properties speed up computations in RNS and provide a high level of design modularity and scalability.

One of the most interesting developments has been the applications of RNS in cryptography [3]. Some cryptographic algorithms which need big word lengths ranging from 2048 bits to 4096 bits like RSA (Rivest-Shamir-Adleman) algorithm [4], have been implemented in RNS [5,6]. RNS is also an appealing method in Elliptic Curve Cryptography (ECC) where the sizes range from 160 to 256 bits.

The modular reduction is the core function in public key cryptosystems, where all calculations are done in a finite field with characteristic $p$.

The first RNS modular reduction proposed by Karl and Reinhard Posch [7] in 1995 was based on the Montgomery reduction algorithm [8]. Their proposed algorithm needed two RNS base extension operations. They used a floating point computation for correction of the base extension in their architecture that was not compatible with the RNS representation.

The main advantage of the RNS Montgomery reduction method is its efficiency in using hardware resources. In this algorithm, half of the RNS channels are involved at a time. Two base extension operations are used to retrieve the other half of the RNS set. The base extension is a costly operation and limits the speed of the algorithm. In 1998, Bajard et al. [9] introduced a new Montgomery RNS reduction architecture using mixed radix system (MRS) [3] representation for base extensions. Due to

the recursive nature of MRS, this method is hard to implement in the hardware. Based on Shenoy and Kumaresan work in [10], Bajard et al. proposed a Montgomery RNS modular reduction algorithm in 1998, using residue recovery for the base extension [11]. In 2000, the floating point approach of [7] was improved by Kawamura et al. [12] by introducing the cox-rower architecture that is well adapted to hardware implementation. In 2014, Bajard and Merkiche [13] proposed an improvement in cox-rower architecture by introducing a second level of Montgomery reduction within each RNS unit. Several variants and improvements on the RNS montgomery modular reduction have been discussed in the literature [2,14–17]. The most recent work in [18] proposed the application of quadratic residues in the RNS Montgomery reduction algorithm.

Modular reduction based on the sum of residues (SOR) algorithm was first presented by Phillips et al. [19] in 2010. The SOR algorithm hardware implementation was proposed later in [20]. A disadvantage of the SOR algorithm is that unlike the Montgomery reduction method, the output is an unknown and variable factor of the "$X \mod p$" value. Although this algorithm offers a high level of parallelism in calculations, the proposed implementation in [20] is considerably big in area.

In this paper, we do an improvement to the sum of residues algorithm by introducing the correction factor $\kappa$ to obtain a precise result. Using an efficient moduli set, we also propose a new design to improve the area in comparison to [20]. The timing of our design is improved compared to RNS Montgomery reduction as well. Two implementations are done for the 256-bit prime field of the SEC2P256K1 [21] and the 255-bit prime field of the ED25519 [22] elliptic curves respectively. It can be extended to other prime fields using the same methodology.

Section 2 of this paper is a quick review of the sum of residues reduction algorithm and the related works already published in the literature. Section 3 is mainly our contribution to the correction of the sum of residues algorithm and improving speed and area using efficient RNS base (moduli set). Section 4 presents our proposed hardware architectures and implementation results. Table A1 in Appendix A summarises the notations applied in this paper.

## 2. Background

Representation of the integer $X$, $0 \leq X < M$, using CRT (Chinese Reminder Theorem) is [3]:

$$X = \big\langle \sum_{i=1}^{N} \langle x_i M_i^{-1} \rangle_{m_i} M_i \big\rangle_M. \tag{1}$$

where, $N$ is the number of moduli in moduli set of co-primes $\mathcal{B} = \{m_1, m_2, \cdots, m_N\}$.

$x_i = X \mod m_i$.

$M = \prod_{i=1}^{N} m_i$ (also called dynamic range of $X$).

$M_i = \frac{M}{m_i}$ and

$M_i^{-1}$ is the multiplicative inverse of $M_i$. In other terms, $M_i \cdot M_i^{-1} \mod m_i = 1$.

We assume that: $2^n > m_1 > \cdots > m_N > 2^{n-1}$. As a result, the dynamic range is $2^{N \cdot (n-1)} < M < 2^{N \cdot n}$.

The values of $M$, $M_i$, and $M_i^{-1}$ are known and pre-computed for hardware implementation.

Consider two $l$-bit integer $X$ and $Y$. The multiplication result $Z = X \cdot Y$ is a $2l$-bit integer. The representation of $Z$ in RNS is :

$$RNS(Z) = \{z_1, z_2, \cdots, z_N\}. \tag{2}$$

where, $z_i = \langle x_i \cdot y_i \rangle_{m_i}$.

The CRT enforces condition $Z < M$. Otherwise, the $N$-tuple RNS set in (2) do not represent the correct integer $Z$. In other terms, the bit length of the moduli ($n$) and the number of moduli ($N$) must be chosen such that the condition $2l < \lceil \log_2 M \rceil < N \cdot n$ is satisfied.

Introducing $\gamma_i = \langle z_i M_i^{-1} \rangle_{m_i}$, the integer $Z$ can be presented as:

$$Z = \langle \sum_{i=1}^{N} \gamma_i M_i \rangle_M. \tag{3}$$

An integer coefficient $\alpha$ can be found such that [3]:

$$Z = \sum_{i=1}^{N} \gamma_i M_i - \alpha M. \tag{4}$$

Reducing $Z$ by the modulus $p$ yields:

$$Z \mod p = \langle Z \rangle_p = \langle \sum_{i=1}^{N} \gamma_i M_i \rangle_p - \langle \alpha M \rangle_p. \tag{5}$$

The RNS multiplications $\langle x_i y_i \rangle_{m_i}$ and $\langle z_i M_i^{-1} \rangle_{m_i}$ can be easily performed by an unsigned integer $n \times n$ multiplier and a modular reduction detailed in Section 2.1. Calculation of $\alpha$ is outlined in Section 2.2.

### 2.1. Efficient RNS Modular Reduction

An RNS modular multiplication in the 256-bit prime field requires a dynamic range of at least 512 bits. In our design this is provided by eight 66-bit pseudo-Mersenne co-prime moduli as presented in Table 1. This moduli set provides a 528-bit dynamic range.

**Table 1.** 66-bit co-prime moduli set $\mathcal{B}$.

| | | | |
|---|---|---|---|
| $2^{66} - 1$ | $2^{66} - 2^2 - 1$ | $2^{66} - 2^3 - 1$ | $2^{66} - 2^4 - 1$ |
| $2^{66} - 2^5 - 1$ | $2^{66} - 2^6 - 1$ | $2^{66} - 2^8 - 1$ | $2^{66} - 2^9 - 1$ |

The optimal RNS bases are discussed in [23]. Modular reduction implementation using moduli set in general form of $m_i = 2^n - 2^{t_i} - 1$ (here $n = 66$) is very fast and low-cost in hardware [24]. Suppose $B$ is a $2n$-bit integer ($0 \leq B < 2^{2n}$). It can be broken up into two $n$-bit most significant and least significant integers denoted as $B_H$ and $B_L$ respectively. In other terms, $B = B_H 2^n + B_L$. Since $\langle 2^n \rangle_{(2^n - 2^{t_i} - 1)} = 2^{t_i} + 1$, then:

$$\langle B_H 2^n + B_L \rangle_{(2^n - 2^{t_i} - 1)} = \langle \langle B_H 2^{t_i} \rangle_{(2^n - 2^{t_i} - 1)} + B_H + B_L \rangle_{(2^n - 2^{t_i} - 1)}. \tag{6}$$

$B_H 2^{t_i}$ has $(n + t_i)$ bits and can be re-written as $B_H 2^{t_i} = B_{HH_i} 2^n + B_{HL_i}$.

Let $(b_{n-1} \ldots b_0)$, $b_i \in \{0, 1\}$ be the binary representation of $B_H$. Then we introduce $B_{HH_i}$ as the most significant $t_i$ bits of $B_H$, i.e. $(b_{n-1} \ldots b_{n-t_i-2})$ and $B_{HL_i}$ as the rest least significant bits $(b_{n-t_i-1} \ldots b_0)$ left shifted $t_i$ times, i.e. $B_{HL_i} = (b_{n-t_i-1} \ldots b_0 \underbrace{0 \cdots 0}_{t_i \text{ zeroes}})$.

Similarly,

$$\langle B_{HH_i} 2^n \rangle_{(2^n - 2^{t_i} - 1)} = \langle B_{HH_i} 2^{t_i} + B_{HH_i} \rangle_{(2^n - 2^{t_i} - 1)}. \tag{7}$$

Since $B_{HH_i}$ is $t_i$ bits long, the term $B_{HH_i} 2^{t_i} + B_{HH_i}$ can be rewritten as concatenation of $B_{HH_i}$ to itself, i.e., $B_{HH_i} 2^{t_i} + B_{HH_i} = B_{HH_i} || B_{HH_i}$. ("||" denotes bit concatenation operation.)

So, the final result is:

$$\langle B \rangle_{(2^n - 2^{t_i} - 1)} = \langle B_{HH_i} || B_{HH_i} + B_{HL_i} + B_H + B_L \rangle_{(2^n - 2^{t_i} - 1)}. \tag{8}$$

The modular reduction of $0 \le B \le 2^{2n}$ can be calculated at the cost of one 4-input $n$-bit CSA (Carry Save Adder) compare to Barrett method [25] that requires two multipliers.

### 2.2. Calculation of $\alpha$

By dividing both sides of (4) to $M$ we obtain:

$$\frac{Z}{M} = \sum_{i=0}^{N} \gamma_i \frac{M_i}{M} - \alpha \to \alpha = \sum_{i=0}^{N} \frac{\gamma_i}{m_i} - \frac{Z}{M}. \tag{9}$$

Since $0 \le \frac{Z}{M} < 1$, then:

$$\alpha = \left\lfloor \sum_{i=0}^{N} \frac{\gamma_i}{m_i} \right\rfloor. \tag{10}$$

It is known that: $0 \le \frac{\gamma_i}{m_i} < 1$, therefore:

$$0 \le \alpha < N. \tag{11}$$

Calculation of $\alpha$ has been discussed in [12,20]. It is shown that choosing proper constants $q$ and $\Delta$ and enforcing boundary condition of (12), $\alpha$ can be calculated using (13).

$$0 \le X < (1 - \Delta)M. \tag{12}$$

$$\alpha = \left\lfloor \frac{1}{2^q} \left( \sum_{i=1}^{N} \left\lfloor \frac{\gamma_i}{2^{n-q}} \right\rfloor + 2^q . \Delta \right) \right\rfloor. \tag{13}$$

Algorithm 1 is used to calculate the coefficient $\alpha$. A Maple program was written to find the optimal $q$ and $\Delta$. Choosing $q = 8$, for the 66-bit moduli set in Table 1, we realised that $\Delta = \frac{1}{2^4}$ is suitable for the whole effective dynamic range. The hardware realisation of (13) is an $N$-input $q$-bit adder with offset $2^q \cdot \Delta$. Note that the effective dynamic range by applying boundary condition in (12), is $\hat{M} = \lfloor (1 - \frac{1}{2^4})M \rfloor$; that is greater than the minimum required bit length (512 bits).

---

**Algorithm 1:** Calculation of $\alpha$

    **input** : $\{\gamma_1, \dots \gamma_N\}$ where, $\gamma_i = \langle z_i \cdot M_i^{-1} \rangle_{m_i}$ , $i \in \{1, \dots, N\}$.
    **input** : $q, \Delta$.
    **output**: $\alpha$.

    $A \leftarrow 2^q \cdot \Delta$ ;
    **for** $i = 1$ **to** $N$ **do**
        $\left| \quad A \leftarrow A + \left\lfloor \frac{\gamma_i}{2^{n-q}} \right\rfloor \right.$ ;
    **end**
    $\alpha \leftarrow \left\lfloor \frac{A}{2^q} \right\rfloor$ ;

---

## 3. Improved Sum of Residues (SOR) Algorithm

Here, we introduce an integer $V$ as:

$$V = \sum_{i=1}^{N} \gamma_i \langle M_i \rangle_p - \langle \alpha M \rangle_p. \tag{14}$$

Comparing (5) and (14), it can be realised that the difference is a factor of modulus $p$. Recalling the fact that for any integer $Z$ we can find an integer $\kappa$ such that $\langle Z \rangle_p = Z - \kappa \cdot p$.

$$\begin{aligned}
V - \langle Z \rangle_p &= \sum_{i=1}^{N} \gamma_i \langle M_i \rangle_p - \langle \sum_{i=1}^{N} \gamma_i M_i \rangle_p \\
&= \sum_{i=1}^{N} \gamma_i (M_i - \nu \cdot p) - \sum_{i=1}^{N} \gamma_i M_i - \mu \cdot p \\
&= (\sum_{i=1}^{N} \gamma_i . \nu - \mu) \cdot p \\
&= \kappa \cdot p.
\end{aligned} \tag{15}$$

($\nu$ and $\mu$ are constants such that: $\langle M_i \rangle_p = M_i - \nu \cdot p$, and $\langle \sum_{i=1}^{N} \gamma_i M_i \rangle_p = \sum_{i=1}^{N} \gamma_i M_i - \mu \cdot p$ ).

The factor ($\kappa$) is a function of $\gamma_i$, not a constant. Therefore the value of $V$ — which is actually the output of SOR algorithm introduced in [19,20] — is not presenting the true reduction of $\langle Z \rangle_p$. In fact:

$$V = \kappa \cdot p + \langle Z \rangle_p = \sum_{i=1}^{N} \gamma_i \langle M_i \rangle_p + \langle -\alpha M \rangle_p. \tag{16}$$

The values of $\langle M_i \rangle_p$ and $\langle \alpha M \rangle_p$ for $\alpha \in \{0, 1, \cdots, N-1\}$ are known and can be implemented in hardware as pre-computed constants.

The RNS form of $V$ resulted from (14) is:

$$\begin{bmatrix} \langle V \rangle_{m_1} \\ \langle V \rangle_{m_2} \\ \vdots \\ \langle V \rangle_{m_N} \end{bmatrix} = \left( \sum_{i=1}^{N} \gamma_i \begin{bmatrix} \langle \langle M_i \rangle_p \rangle_{m_1} \\ \langle \langle M_i \rangle_p \rangle_{m_2} \\ \vdots \\ \langle \langle M_i \rangle_p \rangle_{m_N} \end{bmatrix} \right) + \begin{bmatrix} \langle -\alpha \langle M \rangle_p \rangle_{m_1} \\ \langle -\alpha \langle M \rangle_p \rangle_{m_2} \\ \vdots \\ \langle -\alpha \langle M \rangle_p \rangle_{m_N} \end{bmatrix}. \tag{17}$$

If (17) deducted by $\{ \langle \kappa \cdot p \rangle_{m_1}, \langle \kappa \cdot p \rangle_{m_2}, \cdots, \langle \kappa \cdot p \rangle_{m_N} \}$, the accurate value of $Z_p$ in RNS will be obtained.

$$\begin{bmatrix} \langle Z_p \rangle_{m_1} \\ \langle Z_p \rangle_{m_2} \\ \vdots \\ \langle Z_p \rangle_{m_N} \end{bmatrix} = \left( \sum_{i=1}^{N} \gamma_i \begin{bmatrix} \langle \langle M_i \rangle_p \rangle_{m_1} \\ \langle \langle M_i \rangle_p \rangle_{m_2} \\ \vdots \\ \langle \langle M_i \rangle_p \rangle_{m_N} \end{bmatrix} \right) + \begin{bmatrix} \langle -\alpha \langle M \rangle_p \rangle_{m_1} \\ \langle -\alpha \langle M \rangle_p \rangle_{m_2} \\ \vdots \\ \langle -\alpha \langle M \rangle_p \rangle_{m_N} \end{bmatrix} - \begin{bmatrix} \langle \kappa . p \rangle_{m_1} \\ \langle \kappa . p \rangle_{m_2} \\ \vdots \\ \langle \kappa . p \rangle_{m_N} \end{bmatrix}. \tag{18}$$

### 3.1. Calculation of $\kappa$

Dividing two sides of (16) by $p$ yields:

$$\kappa + \frac{\langle Z \rangle_p}{p} = \sum_{i=1}^{N} \frac{\gamma_i \langle Mi \rangle_p}{p} + \frac{\langle -\alpha M \rangle_p}{p}. \tag{19}$$

The coefficient $\kappa$ is an integer. Reminding that $\frac{\langle Z \rangle_p}{p} < 1$ and $\frac{\langle -\alpha M \rangle_p}{p} < 1$, $\kappa$ can be calculated as:

$$\kappa = \left\lfloor \sum_{i=1}^{N} \frac{\langle \gamma_i M_i \rangle_p}{p} \right\rfloor. \tag{20}$$

The modulus $p$ is considered to be a pseudo Mersenne prime in general form of $p = 2^W - \epsilon$ where $2^W \gg \epsilon$. For example: $p_S = 2^{256} - 2^{32} - 2^9 - 2^8 - 2^7 - 2^6 - 2^4 - 1$ and $p_E = 2^{255} - 19$ are the field modulus for the NIST recommended curve SECP256K1 [21] and the Twisted Edwards Curve ED25519 [22], respectively.

Substitution of fractional equation $\frac{1}{2^W - \epsilon} = \frac{1}{2^W}(1 + \frac{\epsilon}{2^W - \epsilon})$ in (20) results:

$$\kappa = \left\lfloor \sum_{i=1}^{N} \frac{\gamma_i \langle M_i \rangle_p}{2^W} \left( 1 + \frac{\epsilon}{(2^W - \epsilon)} \right) \right\rfloor. \tag{21}$$

Considering that $\langle M_i \rangle_p < p$ and $\gamma_i < 2^n$, if we choose:

$$\epsilon < \frac{2^{W-n}}{N} \tag{22}$$

Then, $\sum\limits_{i=1}^{N} \frac{\gamma_i \langle M_i \rangle_p}{2^W} \frac{\epsilon}{(2^W - \epsilon)} < 1$, and the value of $\kappa$ resulted from (21) is:

$$\kappa = \left\lfloor \sum_{i=1}^{N} \frac{\gamma_i \langle M_i \rangle_p}{2^W} \right\rfloor. \tag{23}$$

The condition in (22) provides a new boundary for choosing the field modulus $p$. It is a valid condition for most known prime modulus $p$ used practically in cryptography. Table 2 shows the validity of $\kappa$ for some standard curves based on (23).

**Table 2.** Checking validity of $\kappa$ for some standard curves [21,22].

| CURVE | Modulus $p$ | $N$ | $n$ | $\frac{2^{W-n}}{N}$ | $\epsilon$ |
|-------|-------------|-----|-----|---------------------|------------|
| ED25519 | $2^{255} - 19$ | 8 | 66 | $2^{186}$ | 19 |
| SECP160K1 | $2^{160} - 2^{32} - 21389$ | 5 | 66 | $\frac{2^{94}}{5}$ | $2^{32} + 21389$ |
| SECP160R1 | $2^{160} - 2^{32} - 1$ | 5 | 66 | $\frac{2^{94}}{5}$ | $2^{32} + 1$ |
| SECP192K1 | $2^{192} - 2^{32} - 4553$ | 6 | 66 | $\frac{2^{125}}{3}$ | $2^{32} + 4553$ |
| SECP192R1 | $2^{192} - 2^{64} - 1$ | 6 | 66 | $\frac{2^{125}}{3}$ | $2^{64} + 1$ |
| SECP224K1 | $2^{224} - 2^{32} - 6803$ | 7 | 66 | $\frac{2^{158}}{7}$ | $2^{32} + 6803$ |
| SECP224R1 | $2^{224} - 2^{96} + 1$ | 7 | 66 | $\frac{2^{158}}{7}$ | $2^{96} - 1$ |
| SECP256K1 | $2^{256} - 2^{32} - 977$ | 8 | 66 | $2^{187}$ | $2^{32} + 977$ |
| SECP2384R1 | $2^{384} - 2^{128} - 2^{96} + 2^{31} - 1$ | 12 | 66 | $\frac{2^{316}}{3}$ | $2^{128} + 2^{96} - 2^{31} + 1$ |
| SECP521R1 | $2^{521} - 1$ | 16 | 66 | $2^{451}$ | 1 |

The hardware implementation of (23) needs a $66 \times 256$-bit multiplier. For an efficient hardware implementation, it is essential to avoid such a big multiplier. To compute the value of $\kappa$ in hardware, we used:

$$\kappa = \left\lfloor \frac{1}{2^T} \sum_{i=1}^{N} \gamma_i \left\lfloor \frac{\langle M_i \rangle_p}{2^{W-T}} \right\rfloor \right\rfloor . \tag{24}$$

The integer $T$ must be selected such that the equality of (23) and (24) is guaranteed. Using a MAPLE program, we realised that $T = 72$ for SECP256K1 and $T = 71$ for ED25519 are the best solutions for an area efficient hardware. In this case, as the term $\left\lfloor \frac{\langle M_i \rangle_p}{2^{W-T}} \right\rfloor$ is 55 bits for SECP256K1 and 44 bits for ED25519, the $66 \times 55$-bit and $66 \times 44$-bit multipliers are required to compute $\kappa$, respectively. Therefore, the coefficient of $\kappa$ for SECP256K1 can be calculated efficiently by the following equation:

$$\kappa = \left\lfloor \frac{1}{2^{72}} \sum_{i=1}^{N} \gamma_i \left\lfloor \frac{\langle M_i \rangle_{p_S}}{2^{184}} \right\rfloor \right\rfloor . \tag{25}$$

Similarly, for ED25519, $\kappa$ can be calculated using the below formula:

$$\kappa = \left\lfloor \frac{1}{2^{71}} \sum_{i=1}^{N} \gamma_i \left\lfloor \frac{\langle M_i \rangle_{p_E}}{2^{184}} \right\rfloor \right\rfloor . \tag{26}$$

The value of $\left\lfloor \frac{\langle M_i \rangle_p}{2^{184}} \right\rfloor$ can be pre-computed and saved in the hardware for $i = 1$ to $N$. The integer $\kappa$ is maximum 52-bit long for SECP256K1 and 42-bit long for ED25519. As a result, RNS conversion is not required. ($\kappa_i = \kappa \mod m_i = \kappa$) and $\kappa$ can be directly used in RNS calculations.

Calculation of $\kappa$ can be done in parallel and will not impose extra delay in the design. Finally, to find $z = X \mod p$ we need to compute $\kappa \cdot \langle p \rangle_{m_i}$. Note that $\langle p \rangle_{m_i}$ is a constant and can be pre-computed as well. So, we get:

$$z_i = \left\langle \langle V \rangle_{m_i} - \langle \kappa \cdot p \rangle_{m_i} \right\rangle_{m_i} . \tag{27}$$

The number of operations can be reduced by pre-computing $\langle -p \rangle_{m_i}$ instead of $\langle p \rangle_{m_i}$.

(A modular subtraction consists of two operations: $\forall a, b < m_i$, $\langle a - b \rangle_{m_i} = \langle a + (m_i - b) \rangle_{m_i}$). Then $z_i$ is calculated directly by:

$$z_i = \left\langle \langle V \rangle_{m_i} + \langle \kappa \cdot \langle -p \rangle_{m_i} \rangle_{m_i} \right\rangle_{m_i} . \tag{28}$$

Algorithm 2 presents the RNS modulo $p$ multiplication $\{x_1, x_2, \cdots, x_N\} \times \{y_1, y_2, \cdots, y_N\}$ mod $p$ over moduli base $\mathcal{B}$ using improved sum of residues method. The calculations at stages 4 and 5 are done in parallel. Different levels of parallelism can be achieved in hardware by adding on or more RNS multipliers to perform stage 5.3 calculations in a shorter time.

As discussed, the coefficient $\kappa$ is a 52-bit(42-bit) integer for SECP256K1(ED25519) design. Consequently, the output of the original SOR algorithm [19] represented in (16) is as big as 308(297) bits. In conclusion, the hardware introduced in [20,26,27] cannot calculate two tandem modular multiplications while the product of the second stage inputs has a higher bit number than the dynamic range that violates the CRT. In cryptographic applications, it is generally required to do multiple

modular multiplications. Our correction to the SOR algorithm ensures that the inputs of the next multiplication stage are in range.

---

**Algorithm 2:** Improved Sum of residues reduction

---

**Require**: $p, \Delta, q, \mathcal{B} = \{m_1, \cdots, m_N\}, m_1 > m_2 > \cdots > m_N, n = \lceil \log_2 m_1 \rceil,$
$W = \lceil \log_2 p \rceil, T, N \geq \lceil \frac{2W}{n} \rceil$

**Require**: $M = \prod\limits_{i=1}^{N} m_i, \hat{M} = (1 - \Delta)M, M_i = \frac{M}{m_i}$ for $i = 1$ to $N$

**Require**: pre-computed tables
$\begin{bmatrix} \langle M_1{}^{-1} \rangle_{m_1} \\ \langle M_2{}^{-1} \rangle_{m_2} \\ \vdots \\ \langle M_N{}^{-1} \rangle_{m_N} \end{bmatrix}$,
$\begin{bmatrix} \langle -p \rangle_{m_1} \\ \langle -p \rangle_{m_2} \\ \vdots \\ \langle -p \rangle_{m_N} \end{bmatrix}$, and
$\begin{bmatrix} \left\lfloor \frac{\langle M_1 \rangle_p}{2^{W-T}} \right\rfloor \\ \vdots \\ \left\lfloor \frac{\langle M_N \rangle_p}{2^{W-T}} \right\rfloor \end{bmatrix}$

**Require**: pre-computed table
$\begin{bmatrix} \langle \langle M_i \rangle_p \rangle_{m_1} \\ \langle \langle M_i \rangle_p \rangle_{m_2} \\ \vdots \\ \langle \langle M_i \rangle_p \rangle_{m_N} \end{bmatrix}$ for $i = 1$ to $N$.

**Require**: pre-computed table
$\begin{bmatrix} \langle \alpha \cdot \langle -M \rangle_p \rangle_{m_1} \\ \langle \alpha \cdot \langle -M \rangle_p \rangle_{m_2} \\ \vdots \\ \langle \alpha \cdot \langle -M \rangle_p \rangle_{m_N} \end{bmatrix}$ for $\alpha = 1$ to $N - 1$

**input** : Integers $X$ and $Y$, $0 \leq X, Y < \hat{M}$ in form of RNS: $\{x_1, \cdots, x_N\}$ and $\{y_1, \cdots, y_N\}$.
**output**: Presentation of $Z = X \cdot Y \mod p$ in RNS: $\{z_1, \cdots, z_N\}$.

1. **for** $i = 1$ **to** $N$ **do**
$\quad | \quad xy_i \leftarrow \langle x_i \cdot y_i \rangle_{m_i}$.
**end**

2. **for** $i = 1$ **to** $N$ **do**
$\quad | \quad \gamma_i \leftarrow \langle xy_i \langle M_i{}^{-1} \rangle_{m_i} \rangle_{m_i}$.
**end**

3. **for** $i = 1$ **to** $N$ **do**
$\quad$ **for** $j = 1$ **to** $N$ **do**
$\quad\quad | \quad Y_{ij} \leftarrow \gamma_i \langle \langle M_i \rangle_p \rangle_{m_j}$.
$\quad$ **end**
**end**

4. **for** $i = 1$ **to** $N$ **do**
$\quad$ 4.1 $\alpha \leftarrow \left\lfloor \frac{1}{2^q} \left( \sum\limits_{i=1}^{N} \left\lfloor \frac{\gamma_i}{2^{n-q}} \right\rfloor + 2^q \Delta \right) \right\rfloor$.
$\quad$ 4.2 $\kappa \leftarrow \left\lfloor \frac{1}{2^T} \sum\limits_{i=1}^{N} \gamma_i \left\lfloor \frac{\langle M_i \rangle_p}{2^{W-T}} \right\rfloor \right\rfloor$.
**end**

5. **for** $i = 1$ **to** $N$ **do**
$\quad$ 5.1 Calculate $\langle \kappa \cdot \langle -p \rangle_{m_i} \rangle_{mi}$.
$\quad$ 5.2 Read $\langle \alpha \langle -M \rangle_p \rangle_{m_i}$ from the table.
$\quad$ 5.3 $sum_i \leftarrow \langle \sum\limits_{j=1}^{N} Y_{ji} \rangle_{m_i}$.
**end**

6. **for** $i = 1$ **to** $N$ **do**
$\quad | \quad z_i \leftarrow \langle sum_i + \alpha \langle -M \rangle_p \rangle_{m_i} + \langle \kappa \langle -p \rangle_{m_i} \rangle_{m_i}$.
**end**

---

## 4. New SOR Algorithm Implementation and Performance

The required memory to implement pre-computed parameters of Algorithm 2 is $N((2N+2)n+n')$ bits, where $n'$ is the biggest bit number of $\left\lfloor \frac{\langle M_i \rangle_p}{2^{W-T}} \right\rfloor, i \in \{1 \cdots N\}$. In our case $n' = 55$ for SECP256K1 and $n' = 44$ for ED25519. Therefore, the required memory is 9944 and 9856 bits for the SECP256K1 and ED25519 respectively.

In our design, FPGA DSP modules are used for the realisation of eight $66 \times 66$ bit multipliers that are followed by a combinational reduction logic to build an RNS multiplier. The total number of 128 DSP resources are used for an RNS multiplier. Table 3 lists maximum logic and net delays of the RNS multiplier and the RNS adder(accumulator) implemented on the different FPGA platforms used in this survey. These delays determine the overall design latency and performance. The maximum RNS adder logic and routing latency are less than half of the RNS multiplier logic and net delays. The system clock cycle is chosen such that an RNS addition is complete in one clock period and an RNS multiplication result is ready in two clock periods.

**Table 3.** Implementation results of SOR components on different FPGAs.

| Unit | Device | Max. Logic Delay (ns) | Max. Net Delay (ns) | Max Achieved Freq. on Core MHz |
|---|---|---|---|---|
| RNS Multiplier | ARTIX 7 | 16.206 | 5.112 | 109.00 |
| RNS Adder | ARTIX 7 | 6.017 | 2.303 | 109.00 |
| RNS Multiplier | VIRTEX 7 | 11.525 | 3.793 | 125.00 |
| RNS Adder | VIRTEX 7 | 3.931 | 1.469 | 125.00 |
| RNS Multiplier | VIRTEX UltraScale+ | 5.910 | 4.099 | 185.18 |
| RNS Adder | VIRTEX UltraScale+ | 2.139 | 2.454 | 185.18 |
| RNS Multiplier | KINTEX 7 | 11.964 | 4.711 | 116.27 |
| RNS Adder | KINTEX 7 | 4.613 | 1.599 | 116.27 |
| RNS Multiplier | KINTEX UltraScale+ | 5.789 | 4.099 | 187.13 |
| RNS Adder | KINTEX UltraScale+ | 2.018 | 2.454 | 187.13 |

Figure 1 presents a simplified block diagram of the Algorithm 2 with non-pipe-lined architecture. We name this architecture as SOR_1M_N. The sequencer state machine provides select signals of the multiplexers and clocks for internal registers. The inputs of the circuit are two 256-bit integers $X$ and $Y$ in RNS representation over base $\mathcal{B}$ ; i.e., $\{x_1, \cdots, x_N\}$ and $\{y_1, \cdots, y_N\}$ respectively.

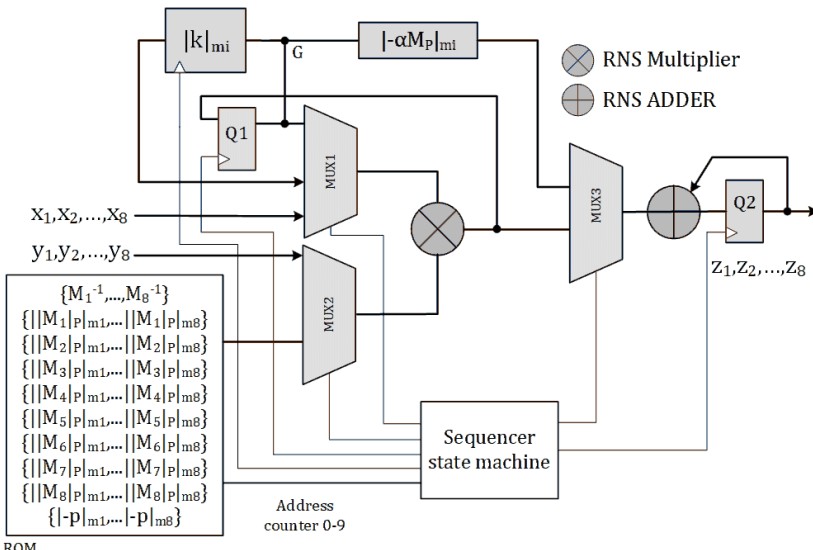

**Figure 1.** Sum of residues reduction block diagram non-pipe-lined (SOR_1M_N) design.

The RNS multiplier inputs are selected by multiplexers MUX1 and MUX2. At the second clock cycle the output of multiplier i.e., $xy_i = \langle x_i \cdot y_i \rangle_{m_i}$ is latched by register $Q1$. At the fourth clock cycle, $\gamma_i = \langle xy_i \cdot M_i^{-1} \rangle_{m_i}$ is calculated and latched by register $Q1$. The calculation of $\alpha$ starts after the fourth clock cycle, by adding the eight most significant bits of $\gamma_1$ to $\gamma_8$ to the offset $2^q \Delta = 2^4$. The 3 most significant bits of the result are used to select the value of $\langle -\alpha \cdot \langle M \rangle_p \rangle_{m_i}$ from the Look up table. Figure 2 illustrates the hardware implementation of $\langle -\alpha \cdot \langle M \rangle_p \rangle_{m_i}$. At the next $3N$ clock cycles $\langle \gamma_i \langle M_i \rangle_p \rangle_{m_j}$ will be calculated and accumulated in register $Q2$. The RNS multiplier must be idle for one clock cycle, letting the RNS adder of the accumulator be completed and latched whenever accumulation of the results is required. The value of $\kappa$ is calculated in parallel using the hardware shown in Figure 3. The $\langle -\kappa p \rangle_{m_i}$ is calculated at the $(3N + 5)$ and $(3N + 6)$ cycles and will be added to the accumulator $Q2$ at the last clock cycle. The sum of moduli reduction is completed in $(3N + 7)$ clock cycles. Figure A1 in appendix A shows the data flow diagram of SOR_1M_N architecture at every clock cycle.

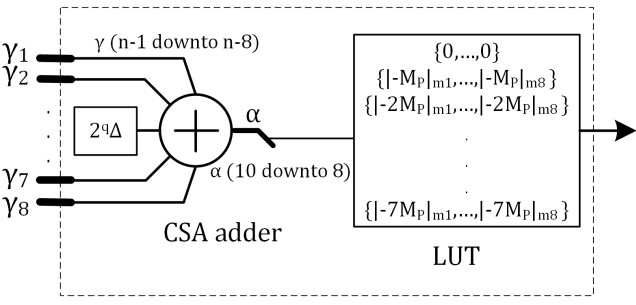

**Figure 2.** Implementation of $\langle \langle -\alpha M \rangle_p \rangle_{m_i}$.

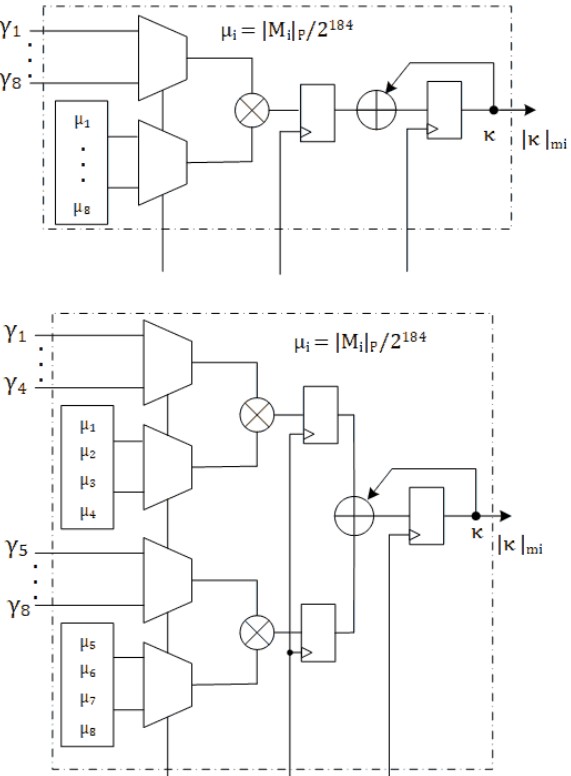

**Figure 3.** Implementation of $\langle \kappa \langle -p \rangle \rangle_{m_i}$ in architectures SOR_1M_N and SOR_1M_P (Up) and in architecture SOR_2M (Down).

A pipe-lined design is depicted in Figure 4. Here, an extra register Q3 latches the RNS multiplier's output. So, The idle cycles in SOR_1M_N are removed. We call this design SOR_1M_P. The data flow diagram of SOR_1M_P architecture is illustrated in Figure A2 in Appendix A. Algorithm 2 can be performed in $2(N+4)$ clock cycles using this architecture.

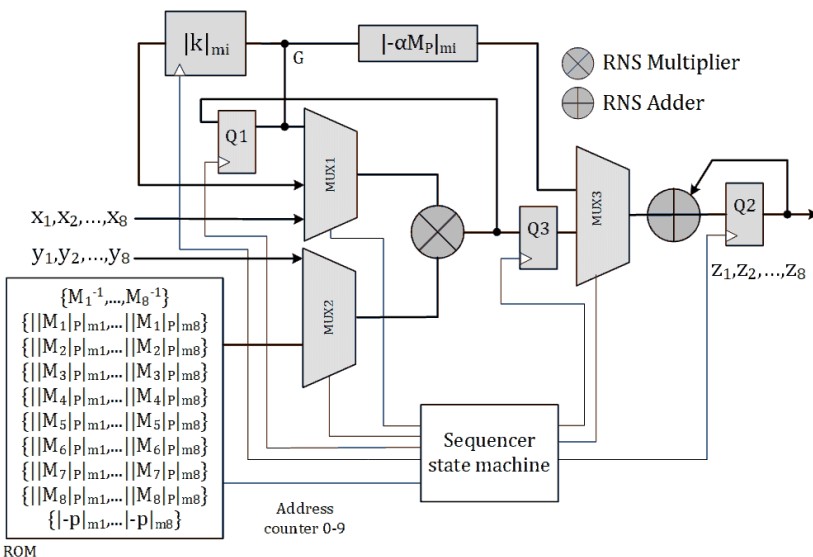

**Figure 4.** Sum of residues reduction block diagram with pipe-lined (SOR_1M_P) design.

Parallel designs are possible by adding RNS multipliers to the design. Figure 5 shows the architecture of using two identical RNS multipliers in parallel to implement algorithm 2. We tag this architecture as SOR_2M. The calculation of $\langle \gamma_i \langle M_i \rangle_p \rangle_{m_j}$, $(i = 1 \cdots N)$ is split between two RNS multipliers. So, the required time to calculate all the $N$ terms is halved. As shown in Figure 3, An extra $n \times n'$ multiplier is also required to calculate $\kappa$ in time. The latency of SOM_2M architecture is $2(\frac{N}{2}+5)$ clock cycles. Theoretically, the latency could be as small as 12 clock cycles using $N$ parallel RNS multipliers. Figure A3 in appendix A shows the data flow diagram of SOM_2M architecture.

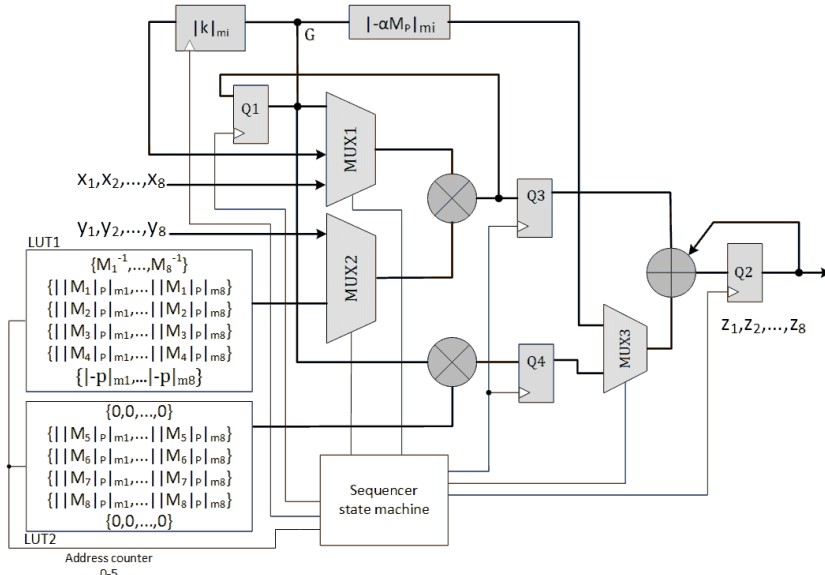

**Figure 5.** Sum of residues block diagram using two parallel pipe-lined (SOR_2M) design.

Table 4, shows implementation results on ARTIX 7, VIRTEX 7, KINTEX 7, VIRTEX UltraScale+™, and KINTEX UltraScale+ FPGA series. VIVADO 2017.4 is used for VHDL codes synthesis. On a Xilinx

VIRTEX 7 platform, as shown in Table 3, a 66-bit modular multiplication was achieved in 11.525 ns and a 66-bit RNS addition was performed in 3.93 ns. Considering the maximum net delays, clock frequency 125 MHz is achievable. The fastest design is realised using KINTEX UltraScale+ that clock frequency 187.13 MHz is reachable. Figure 6 summarises the latency and throughput of SOR_1M_N, SOR_1M_P, and SOR_2M on different Xilinx FPGA series for ease of comparison.

**Table 4.** Sum of residues reduction algorithm Implementation on Xilinx FPGAs.

| Architecture | Platform FPGA | Clk Frequency (MHz) | Latency (ns) | Area (KLUTs),(FFs),(DSPs) | Throughput (Mbps) |
|---|---|---|---|---|---|
| SOR_1M_N | ARTIX 7 | 92.5 | 335 | (8.17),(3758),(140) | 1671 |
| SOR_1M_N | VIRTEX 7 | 128.8 | 241 | (8.17),(3758),(140) | 2323 |
| SOR_1M_N | KINTEX 7 | 117.67 | 263 | (8.29),(3758),(140) | 2129 |
| SOR_1M_N | VIRTEX US+ [1] | 192 | 157 | (8.14),(3758),(140) | 3567 |
| SOR_1M_N | KINTEX US+ | 198 | 156.5 | (8.29),(3758),(140) | 3578 |
| SOR_1M_P | ARTIX 7 | 92.5 | 259.5 | (8.73),(4279),(140) | 2158 |
| SOR_1M_P | VIRTEX 7 | 138.8 | 173 | (8.73),(4279),(140) | 3237 |
| SOR_1M_P | KINTEX 7 | 117.6 | 204 | (8.89),(4279),(140) | 2745 |
| SOR_1M_P | VIRTEX US+ | 185.18 | 130 | (8.71),(4279),(140) | 4307 |
| SOR_1M_P | KINTEX US+ | 187.13 | 128.3 | (8.89),(4279),(140) | 4364 |
| SOR_2M | ARTIX 7 | 92.5 | 194.6 | (10.11),(4797),(280) | 2877 |
| SOR_2M | VIRTEX 7 | 128.5 | 140 | (10.11),(4797),(280) | 3998 |
| SOR_2M | KINTEX 7 | 121.9 | 147.6 | (10.27),(4797),(280) | 3794 |
| SOR_2M | VIRTEX US+ | 185.18 | 97.3 | (10.11),(4797),(280) | 5761 |
| SOR_2M | KINTEX US+ | 187.13 | 96.3 | (10.26),(4797),(280) | 5821 |

[1] US+: Ultra Scale+ ™.

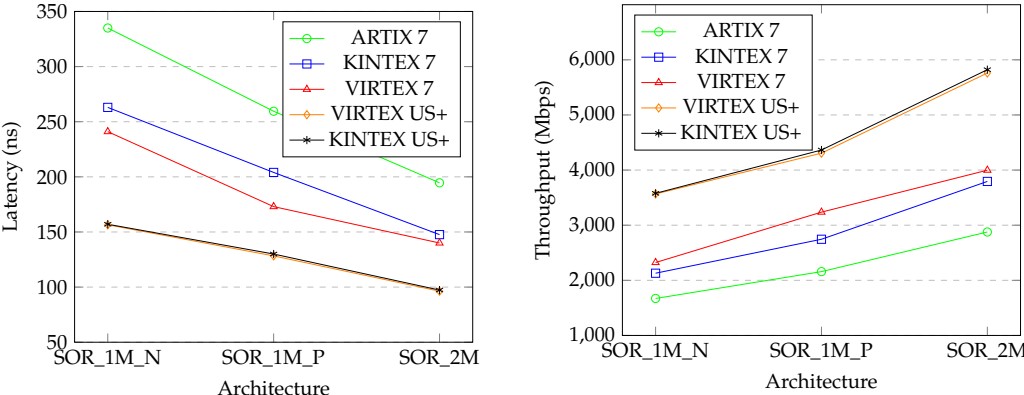

**Figure 6.** SOR architectures Latency and Throughput on Xilinx FPGAs.

*4.1. Comparison*

In Table 5, we have outlined implementation results of recent similar works in the context of RNS. The design in [20] and [27] are based on the SOR algorithm in [19]. Both of them use forty 14-bit co-prime moduli as RNS base to provide a 560-bit dynamic range. Barrett reduction method [25] is used for moduli multiplication at each channel. The Barrett reduction algorithm costs 2 multiplications and one subtraction which is not an optimised method for high-speed designs. The design in [20] is a combinational logic and performs an RNS modular reduction in one clock cycles. The area of this design is reported in [27] which is equivalent to (34.34 KLUTs, 2016 DSPs) for non pipe-lined and (36.5 KLUTs, 2016 DSPs) for pipe-lined architectures. The MM_SPA design in [27], is a more

reasonable design in terms of the logic size (11.43 KLUT, 512 DSPs). However, in contrast to our SOR_2M design on VIRTEX-7, it consumes more hardware resources and in terms of speed, it is considerably slower. These designs, are based on SOR algorithm in [19] that is not performing a complete reduction. As discussed in Section 3.1, their outputs can exceed the RNS dynamic range and give out completely incorrect results.

A survey on RNS Montgomery reduction algorithm and the improvements in this context is presented in [18]. The application of quadratic residues in RNS modular reduction is then presented and two algorithms sQ-RNS and dQ-RNS are proposed. The authors used eight 65-bit moduli base for their RNS hardware which is similar to our design. The achieved clock frequencies for these two designs are 139.5 MHz and 142.7 MHz, respectively. The input considered for the algorithms is the RNS presentation of "$K^2 \cdot x$"; where "$x$" is equivalent to $Z$ in our notations in Equation (2) and "$K^2$" is a constant. To do a fair comparison, it is required to consider two initial RNS multiplications to get the input ready for the algorithms sQ-RNS and dQ-RNS. This adds two stages of full range RNS multiplication to the design.

**Table 5.** Comparison of our design with recent similar works.

| Design | Platform | Clk Frequency (MHz) | Latency (ns) | Area (KLUT),(DSP) | Throughput (Mbps) |
|---|---|---|---|---|---|
| MM_PA_P [20] | VIRTEX 6 | 71.40 | 14.20 | (36.5),(2016) [1] | 14798 |
| MM_PA_N [20] | VIRTEX 6 | 21.16 | 47.25 | (34.34),(2016) [1] | 5120 |
| MM_PA_P [27] | VIRTEX 7 | 62.11 | 48.3 | (29.17),(2799) | 15900 |
| MM_SPA [27] | VIRTEX 7 | 54.34 | 239.2 | (11.43),(512) | 1391 |
| (Ours) SOR_1M_P | VIRTEX 7 | 138.8 | 173 | (8.73),(140) | 3237 |
| (Ours) SOR_2M | VIRTEX 7 | 128.5 | 140 | (10.11),(280) | 3998 |
| sQ-RNS [2] | KINTEX US+ | 139.5 | 107.53(150.53) | (4.247),(84) | 4835 [2] |
| dQ-RNS [18] | KINTEX US+ | 142.7 | 126.14(168.18) [2] | (4.076),(84) | 4122 [2] |
| (Ours) SOR_1M_P | KINTEX US+ | 187.13 | 128.3 | (8.89),(140) | 4364 |
| (Ours) SOR_2M | KINTEX US+ | 187.13 | 96.3 | (10.26),(280) | 5821 |

[1] Area reported in [27]; [2] Our estimation.

As illustrated on Figure 13 of [18] it takes 3 clock cycles to perform one multiplication and reduction. So, at the maximum working clock frequency, 42 ns will be added to the latency of the proposed RNS modular reduction circuit. As a result, the equivalent latency for an RNS reduction for sQ-RNs and dQ-RNS reduction hardware is 150.53 ns and 168.18 ns, respectively. Consider that the output of these algorithms is a factor of "$\langle x \cdot M^{-1} \rangle_p$", not the precise value of "$\langle x \rangle_p$". The RNS Montgomery reduction algorithms use half of moduli set. This makes the hardware area efficient, but it still full moduli range multiplication are required for computations. On the same FPGA platform used in [18], i.e., KINTEX Ultra Scale+ ™, we achieved the latency of 128.3 ns and 96.3 ns with our SOR_1M_P and SOR_2M designs, respectively. The latency of SOR_2M showed 36% improvement compare to sQ-RNS and 41.1% improvement in contrast to MM_SPA on similar FPGA platforms. Similarly, there is 14.9% and 27.6% improvement of SOR_1M_P latency in compare to sQ-RNS and MM_SPA designs, respectively. The latency of our SOR_M_N, however, is very close to sQ-RNS and MM_SPA designs.

## 5. Conclusions

We introduced a coefficient $\kappa$ to make a correction on the SOR algorithm to compute the precise value of modular reduction directly in Residue Number Systems for application in cryptography. We also proposed three hardware architectures for the new SOR algorithm and implemented them on different FPGA platforms. Comparing our implementation results to recent similar works showed an improvement achieved in terms of the speed. The sum of residues algorithm is naturally modular and can use parallel multipliers to speed up calculations. It fits for applications where high-speed modular calculations are in demand. This algorithm uses more hardware resources in compare to RNS Montgomery reduction method. Variants of the SOR algorithm can be studied in future works to achieve an area efficient hardware.

**Funding:** This research received no external funding

**Conflicts of Interest:** The authors declare no conflict of interest.

## Appendix A

Data flow diagram per clock cycle for the SOR architectures listed in Section 4 are illustrated in Figures A1–A3.

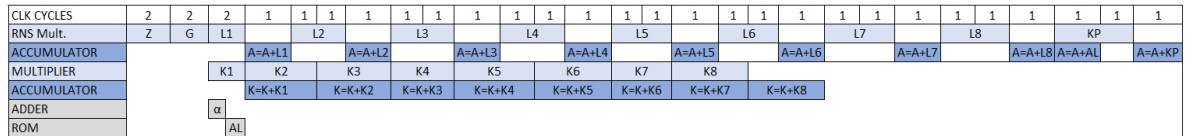

**Figure A1.** Data flow of SOR non pipelined with one RNS multiplier architecture SOR_1M_N.

**Figure A2.** Data flow of SOR with one pipelined RNS multiplier architecture SOR_1M_P.

**Figure A3.** Data flow of SOR with two RNS multiplier architecture SOR_2M.

**Table A1.** The notations applied in this paper.

| Notation | Description |
| --- | --- |
| $p$ | Field modulus. In this work considered as a 256-bit prime $p_S = 2^{256} - 2^{32} - 977$ or 255-bit prime $p_E = 2^{255} - 19$. |
| $m_i$ | RNS channel modulus. $m_i = 2^n - 2^{t_i} - 1$ , $t_i \in \{0, 2, 3, 4, 5, 6, 8, 9\}$. |
| $n$ | Bit-length of modulus $m_i$. ($n = max\lceil log_2 m_i \rceil, i \in \{1, \ldots, N\}$). |
| $n'$ | Is the maximum bit number of $\left\lfloor \frac{\langle M_i \rangle_p}{2^{W-T}} \right\rfloor, i \in \{1 \cdots N\}$. |
| $\mathcal{B}$ | set of RNS Moduli: $\mathcal{B} = \{m_1, m_2, \ldots, m_N\}$. |
| $N$ | Number of moduli in $\mathcal{B}$ (size of $\mathcal{B}$). |
| $B$ | Is a $2n$-bit integer, product of two RNS channels. |
| $B_H$ | Is the $n$ most significant bits of $B$, i.e., $B_H = \left\lfloor \frac{B}{2^n} \right\rfloor$. |
| $B_L$ | Is the $n$ least significant bits of $B$, i.e., $B_L = B \mod 2^n$. |
| $B_{HH_i}$ | Is the $t_i$ most significant bits of $2^{t_i} B_H$, i.e., $B_{HH_i} = \left\lfloor \frac{B_H}{2^{n-t_i}} \right\rfloor$. |
| $B_{HL_i}$ | Is the $n$ least significant bits of $2^{t_i} B_H$, i.e., $B_{HL_i} = 2^{t_i} B_H \mod 2^n$. |
| $A$ | denotes accumulator in Algorithm 1 and Figures A1–A3. |
| $X, Y$ | Integers that meet the condition $0 \le X \cdot Y < M$. |
| $Z$ | An integer considered as product of $X$ and $Y$. |
| $x_i$ | The residue of integer $X$ in channel $m_i$ i.e., $x_i = X \mod m_i$. |
| $\langle Z \rangle_p$ | Mod operation $Z \mod p$. |
| $RNS(X)$ | The RNS function. Returns the RNS representation of integer $X$. |
| $\{x_1, x_2, \ldots, x_N\}$ | RNS representation of integer $X$. |
| $\begin{bmatrix} x_1 \\ x_2 \\ \vdots \\ x_n \end{bmatrix}$ | RNS representation of integer $X$. |
| $(b_{n-1} b_{n-1} \ldots b_0)$ | Binary representation of an $n$-bit integer $B$. ($b_i \in \{0, 1\}$). |
| $\|$ | Bit concatenation operation. |
| $\lceil u \rceil$ | The function $ceil(u)$. |
| $\lfloor u \rfloor$ | The function $floor(u)$. |
| $W$ | Bit-length of modulus $p$, i.e., $W = \lceil log_2 p \rceil$. |
| $M$ | The dynamic range of RNS moduli. $M = \prod\limits_{i=1}^{N} m_i$. |
| $M_i$ | Is defined as $M_i = \frac{M}{m_i}$. |
| $\hat{M}$ | Is the effective dynamic range. $\hat{M} = M(1 - \Delta)$. |
| $\Delta$ | Correction factor used to calculate $\alpha$. In our design $\Delta = \frac{1}{2^4}$. |
| $Gi$ | Is: $\gamma_i = \langle z_i \cdot M_i^{-1} \rangle_{m_i}, i \in \{1, \ldots, 8\}$. |
| $Li$ | Is: $\{ \langle Gi \cdot \langle M_i \rangle_p \rangle_{m_1}, \ldots, \langle Gi \cdot \langle M_i \rangle_p \rangle_{m_N} \}$. |
| $Ki$ | Is: $\left\lfloor \frac{1}{2^T} \gamma_i \left\lfloor \frac{\langle M_i \rangle_p}{2^{W-T}} \right\rfloor \right\rfloor$. |
| $K$ | Is the $\kappa$ accumulator. |
| $AL$ | Is: $\{ \langle \alpha \cdot \langle -M \rangle_p \rangle_{m_1}, \ldots, \langle \alpha \cdot \langle -M \rangle_p \rangle_{m_N} \}$. |
| $KP$ | Is: $\{ \kappa \cdot (-p_1), \ldots, \kappa \cdot (-p_N) \}$. |

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
