# Peer review of "Improved Sum of Residues Modular Multiplication Algorithm"

_cryptography, doi:10.3390/cryptography3020014_

Round 1
Reviewer 1 Report
1. The redaction of multiple sentences should be reviewed; in most of the cases a word is missing, in others the idea is not clear.
These include, but are not limited to lines: 24, 28, 37, 40, 41, 47, 55, 72, 73, 86, 93, 101, 107, 110, 112, 113, 116, 123, 124, 127, 134-136, 148, 170-173, 200, 210.
2. The discussion from line 35 to line 45 could be omitted since it discusses Montgomery RNS, which is not the focus of the paper.
3. The paragraph that begins in line 55 should be split after "methodology." The first part can be appended to the previous paragraph, and the second part should be on its own.
4. Should use the command \cdot to represent multiplication rather than a regular dot.
5. Line 73: single capitals like $n$ and $N$ can be enclosed in parenthesis to avoid using too many commas.
6. Lines 55, 79: "a/an n-bit prime field" or "the n-bit prime field."
7. Line 79: "low cost" -> "low-cost"
8. Line 79: "is $2n$-bit integer" -> "is a $2n$-bit integer"
9. Line 81: The || symbol is most commonly used to represent concatenation in hardware design.
10. I think the right side of Eq. 8 can be exempted from using modulo notation since it is implied that the statement between angles is the reduction of A mod p. The same occurs on other equations, please review.
11. Algorithm 1:
a) Consider stating 1st input as: "$[\gamma_1 \ldots \gamma_N]$"
Consider adding to 1st input: "where $\gamma_i = \{y_{n-1} \ldots y_{n-q} \ldots y_0\}$"
b) Consider stating 1. as: "$A \leftarrow 2^q \cdot \Delta$"
Consider adding to 1.: "where $A = \{a_{n-1} \ldots a_{n-r} \ldots a_0\}$"
c) The package you are employing for writing algorithms should support the For structure.
If you are using algorithmic try for 2.:
\FOR{i=1 \TO N}
\STATE statement
\ENDFOR
d) Consider writing the for statement as: "$A \leftarrow A + \{y_{n-1} \ldots y_{n-q}\}"
e) Consider adding before 3.: "$r = \lceil \log_2 N \rceil$"
Consider stating 3. as: "$\alpha \leftarrow \{a_{n-1} \ldots a_{n-r}\}$"
12. Line 95: I think the mod notation is missing its angles.
13. Review third line in Eq. 15.
14. Line 97:
a) ", ($k$)," -> " ($k$)"
b) "of $\gamma_i$ not" -> "of $\gamma_i$, not"
c) Use --- to get the correct punctuation mark
15. Line 102: I believe it is not necessary to spell "equation" when you are using \eqref.
16. Eq. 21: Consider employing the $\left( x \right)$ construction so that the resulting parenthesis are of the appropriate size.
17. Algorithm 2: If you choose to update Algorithm 1, this should be updated as well.
18. Figure 3 is not referenced in the paper.
19. Line 160: Figure 3 should be referenced here.
20. Line 173: Table 4 was introduced at the beginning of the paragraph, last sentence can be removed.
21. Line 175: "table" -> "Table"
22. Table 4: Area should be provided in terms of SLC, FF, LUT, DSP, and BRAM (if used).
23. If you are concerned with the logic size as implied in line 183, you should provide results for an Artix FPGA. These are more resource-friendly than Virtex/Kintex fabrics albeit slower.
Moreover, the results reported in this work can be enriched by providing power estimation figures. However, Vivado per default has a resolution of Watts so I'm not sure if it would produce any significant variation for your designs.
24. Reference 4. "rns" -> "RNS"
25. The comparison with the state of the art works would be more clear if you use some graphs.
26. (Important) At first glance I don't see the merit of your designs when compared against the State of the Art in terms of Throughput/Area. Might want to remark their advantages in some way.
Author Response
1. Noted. Correction done.
2. We prefer to keep the discussion on RNS Montgomery reduction (lines 35 to 45) for two reasons. First, to discuss the developments in RNS reduction methods before 2010 that SOR introduced. Second, to show the road map to developed RNS MMR discussed in [16]. Then, we used the results of [16] to show that SOR can be notably faster on the same platform at the cost of larger area. If it is preferred to omit the RNS Montgomery, then it is better to delete the comparison to [16] as well.
3. Done
4. \cdot considered for all multiplication signs.
5. Corrected.
6. Corrected.
7. Corrected.
8. Corrected.
9. Symbol “&” is used in VHDL for concatenation. Changed to “||” as advised.
10. The right side of Eq. 8 is sum of four integers. Three of these integers are n-bit. So, the result can be bigger than 2.m_i. The sum of three n-bit numbers may be needed to be subtracted by 2.m_i or m_i to get the correct reduction. So we have to keep the angles. In the hardware, the output of the adder is compared with 2.m_i and m_i and a subtraction is done if needed to ensure the output is less than m_i.
11. Package Algorithm2e is used and the advised changed considered for both Algorithm 1 and Algorithm 2. Advised corrections done.
12. Mod notation added.
13. Corrected.
14. All points corrected.
15. “equation” removed.
16. Corrected.
17. Algorithm 2 updated for package Algorithm2e
18. Reference corrected at line 160.
19. Corrected.
20. Corrected.
21. Corrected.
22. Number of FF provided in Table 4. No SLC or BRAM used in this design.
23. The focus of this design and RNS application is on the speed. There is no major difference in power consumption for different designs of this work. Moreover, Power consumption is not addressed in the referenced papers and cannot have a comparison.
Xilinx series 7 family including Artix, Virtex and Kintex -7 have a similar architecture design. There wouldn’t be much difference in synthesised area on these devices. The target application for VIRTEX-7 devices is high performance computing. Used in most of the RNS implementations in the literature and can gives us a chance to have a fair comparison. Kintex Ultra Scale + was used in [16]. We showed our results on Kintex/Virtex Ultra Scale + devices to have a fair comparison to fastest RNS Montgomery reduction in the literature.
We synthesised our design on ARTIX7. There is no change in area as expected but it is slower. The net delays are:
RNS Multiplier : Max. logic delay : 16.206 ns , Max. net delay: 5.112 ns
RNS Adder : Max. logic delay: 6.017 ns , Max. net delay: 2.303 ns.
The achieved frequency is 92.5 MHz. The SOM_2M design latency would be 194.5ns.
Latency of SOM_1M_N is 335 ns and
Latency of SOM_1M_P is 259.5 ns;
Results added to Table 3 and Table 4.
24. Corrected.
25. Comparing designs performance is possible through Time x Area factor. Unfortunately there is no way to normalise the area. We cannot say a DSP slice is equivalent to how many LUTs for instance and figure out Area as a single figure. We haven’t seen graph presentations in similar papers as well. The Table presentation of LUT/FF/DSP/BRAM used in the design gives a perspective to the reader to decide which design is using less hardware resources.
Graphs for latency and throughput added.
26. Basically, the purpose of this paper is to make a correction on SOR algorithm and propose a new hardware for the improved algorithm discussed at the first half of the paper. We added lines 124-130 to make it clear why the older SOR algorithm will not work correctly in cryptographic applications. Consequently, The hardware implemented based on the old SOR algorithm will not work correctly. There is no sense to compare our design to what is not practical even if it has a better throughput /Area. In terms of Area, the SOR algorithm implementations in the literature are generally bigger than our design. According to Table. 5, designs in [1] and [3] are much bigger than ours using more than 2000 DSPs. Even our SOR_2M design is using less hardware resources. The RNS Montgomery algorithm is using half of the moduli set and the hardware expected to be half of the SOR’s. Although, this is not an apple to apple comparison. On Table 4 and 5, we showed that SOR modular multiplication can be more than 40% faster than RNS Montgomery modular multiplication.
Reviewer 2 Report
The authors present an improved hardware efficient design for SOR for modular applications.
Few suggestions to improve the manuscript:
- A clear comparison of proposed versus existing design of SOR (say improved version of figure 1 by showcasing explicit difference between proposed and existing design) will be helpful.
- As the manuscript involves lot of math equations, having a table explaining different variables used will ease the reading and useful for referencing.
- Comparison results need to be improved. For instance, Table 4 needs to be improved by adding the SOR_1M_N/P with estimates for existing work will be helpful.
- A state-of-the-art section with crisp distinction is missing.
Author Response
1. Basically, the purpose of this paper is to make a correction on SOR algorithm and propose a new hardware for the improved algorithm discussed at the first half of the paper.
We added lines 123-129 to make it clear why the older SOR algorithm will not work correctly in cryptographic applications. Consequently, The hardware implemented based on the old SOR algorithm will not work correctly.
Subsection 4.1 compares our design in terms of Area/Speed and performance with the SOR and Montgomery reduction implementations.
2. Table of notations added on Appendix B. (we moved the notation of graphs at appendix A to this table)
3. Table 4 includes results of SOR_1M_N/P. I think you mean we need to copy these results to Table 5 for ease of comparison. Our design results added to Table 5. Distinct comparison added in section 4.1.
4. We detected a problem in SOR algorithm. The output of the traditional algorithms is not a complete modular reduction. Two tandem modular reduction is not possible as the RNS dynamic range is violated. We proposed a new SOR algorithm with correction factor $\kappa$ that removes this problem.
In section 3, we clearly stated what is the problem of the initial SOR algorithm , Why it is not working and how we did the correction (3.1 Calculation of $\kapp$).
Hardware implementation of the correction
module costs some hardware resources, but as it is done in parallel to other
calculations the timing will not be affected. There is no sense to compare Area/Timing of our design with designs
based on the old SOR algorithm due to clear reason: “The old SOR algorithm It
is not working consequently the hardware based on this algorithm is useless to
implement for applications in cryptography that there is need to perform
multiple modular reductions/multiplications”.
Round 2
Reviewer 1 Report
There are still some typos in the document. Particularly some punctuation marks have been misplaced and spaces are missing around some reference quotations (e.g. lines 24 and 142). A capital also appears to be missing in table A.1.
Figure 3 is still not referenced in the paper.
Your implementation should certainly report an SLC count (for Xilinx an SLICE is a bundle of LUTs and FFs) unless you are providing post-synthesis results. Please clearly state whether the results in Table 4 were obtained post-synthesis or post-PAR.
Author Response
1. Typos corrected in lines 18, 24, 36, 44, 95, 96, 142, 181-183, caption of Figure 2, equation (15) . In line 29 the second “ RNS Montgomery reduction” clause omitted. Line 168, "ARTIX 7" added. Lines 193, and 200 : quotations added to $K^2 \cdot x$ , $x$, $K^2$, $x \cdot M_i$, and $x_p$ as they are notations used in [16].
To avoid mis-interpretation some notations changed: Lines 80 -85: $A$ in equation (6),(7),(8) converted to $B$, (A is used for accumulator in algorithm 1 and Figures A1, A2, and A3.) .
The bit-length of $p$ changed to $W$ , to avoid confusion; as $K$ is used for $\kappa$ accumulator in figures A1 to A3. All $2^K$ and $2^{K-T}$ changed to $2^W$ and $2^{W-T}$ , respectively.
2. Notations $A&, $B$ , $W$ , $B_H$ , $B_L$, $B_{HH_i}$, $B_{HL_i}$ added to Table A1.
3. Reference to Figure 3 corrected in line 163.
4. It is clearly referenced as “implementation of the SOR algorithm on Xilinx FPGAs” on Table 4 that refers to Post partitioning results. Not post-synthesis. Otherwise it would be referenced as “synthesis results”. We used VIVADO version 2017.04 for implementation of our design. Unlike XILINX ISE, VIVADO does not use SLICE count in the utilization reports. The rationale is that Slices can be and often are partially used and that's why slice utilization isn't very indicative of the density of the design.
The 7 -series of Xilinx FPGAs have similar architecture. Every Slice in the 7-family consists of 4 LUTs and 8 FFs. So, the slice utilization is the worst case of (LUT/4) and (FF/8). (whichever is greater ). It can be calculated by the reader for comparison. As VIVADO is not reporting SLICE count directly, and recent papers working with VIVADO have not reported SLICE count as well. We do not use SLICE count as in is not explicitly reported by the firmware we use, persuming that the reader can have a very close estimation by simple calculations.
Reviewer 2 Report
Authors have addressed or clarified in response to my concerns.
I ask the authors to thoroughly proofread for final submission, as some typos still exist.
Author Response
1. Typos corrected in lines 18, 24, 36, 44, 95, 96, 142, 181-183, caption of Figure 2, equation (15) . In line 29 the second “ RNS Montgomery reduction” clause omitted. Line 168, "ARTIX 7" added. Lines 193, and 200 : quotations added to $K^2 \cdot x$ , $x$, $K^2$, $x \cdot M_i$, and $x_p$ as they are notations used in [16].
To avoid mis-interpretation some notations changed: Lines 80 -85: $A$ in equation (6),(7),(8) converted to $B$, (A is used for accumulator in algorithm 1 and Figures A1, A2, and A3.) .
The bit-length of $p$ changed to $W$ , to avoid confusion; as $K$ is used for $\kappa$ accumulator in figures A1 to A3. All $2^K$ and $2^{K-T}$ changed to $2^W$ and $2^{W-T}$ , respectively.
2. Notations $A&, $B$ , $W$ , $B_H$ , $B_L$, $B_{HH_i}$, $B_{HL_i}$ added to Table A1.
3. Reference to Figure 3 corrected in line 163.